

# Embryonic exposure to fentanyl induces behavioral changes and neurotoxicity in zebrafish larvae

Binjie Wang, Jiale Chen, Zhong Sheng, Wanting Lian, Yuanzhao Wu and Meng Liu

The Department of Criminal Science and Technology, Zhejiang Police College, Hangzhou, Zhejiang, China

## ABSTRACT

The use of fentanyl during pregnancy, whether by prescription or illicit use, may result in high blood levels that pose an early risk to fetal development. However, little is known regarding the neurotoxicity that might arise from excessive fentanyl exposure in growing organisms, particularly drug-related withdrawal symptoms. In this study, zebrafish embryos were exposed to fentanyl solutions (0.1, 1, and 5 mg/L) for 5 days post fertilization (dpf), followed by a 5-day recovery period, and then the larvae were evaluated for photomotor response, anxiety behavior, shoaling behavior, aggression, social preference, and sensitization behavior. Fentanyl solutions at 1 and 5 mg/L induced elevated anxiety, decreased social preference and aggressiveness, and behavioral sensitization in zebrafish larvae. The expression of genes revealed that embryonic exposure to fentanyl caused substantial alterations in neural activity (*bdnf*, *c-fos*) and neuronal development and plasticity (*npas4a*, *egr1*, *btg2*, *ier2a*, *vgf*). These results suggest that fentanyl exposure during embryonic development is neurotoxic, highlighting the importance of zebrafish as an aquatic species in research on the neurobehavioral effects of opioids in vertebrates.

## INTRODUCTION

Opioids, a group of alkaloids and derivatives isolated from the opium poppy plant, are frequently used to treat cancer pain, post-operative pain from surgery, and other types of chronic pain. In the past two decades, drug dependence and addiction due to the nonmedical use of opioids have been identified as a significant public health problem (*Coyle et al., 2018*).

Chronic accumulation of toxic opioid metabolites in adults can result in opioid-induced neurotoxicity, which includes hallucinations, delirium, confusion, seizures, and nociceptive hypersensitivity (*vanden Bosch et al., 2017*; *Herlinger & Lingford-Hughes, 2022*). Additionally, the exposure of women to excessive doses of opioids, including from illicit sources, raises the risk to fetal development, which is extremely concerning. From 2000 to 2007, a survey of 1.1 million Medicaid-enrolled women in 46 states and the District of Columbia in the United States revealed that one in five of these women obtained an opioid prescription during pregnancy (*Desai et al., 2014*). Certain birth

Corresponding authors
Binjie Wang, wangbinjie@zjjcxy.cn
Meng Liu, liumeng@zjpc.edu.cn

abnormalities, including congenital heart disease, spinal deformities, hydrocephalus, and glaucoma, have been associated with the use of opioid analgesics in early pregnancy (*Broussard et al., 2011*; *Martins et al., 2019*). Self-reported opioid usage among pregnant women rose 14-fold between 2000 and 2012, contributing to a 400% rise in newborn morbidity (*Winkelman et al., 2018*; *Honein, Boyle & Redfield, 2019*). Prenatal exposure to opioids also caused behavioral changes in infants, including psychomotor developmental impairments, cognitive delays, and cognitive function deficits (*Patrick et al., 2012*; *Conradt et al., 2019*). *Minnes, Lang & Singer (2011)* proposed a significant correlation between anxiety, aggression, feelings of rejection, disruptive or inattentive behavior in newborns and prenatal opioid exposure. However, the neurotoxic implications of early developmental exposure to opioids are still unclear.

The zebrafish (Danio rerio) has been widely employed in genetic and developmental research in recent years (*Bailey, Oliveri & Levin, 2013*). With the varying chemical structures of new psychoactive substances, such as synthetic cannabinoids (*Garcia-Gonzalez et al., 2021*), synthetic opioids (*Kolesnikova et al., 2021*), and hallucinogenic drugs (*Demin et al., 2022*), the zebrafish model provides an efficient and rapid method to study the toxicity assessment and mechanisms of these compounds. Zebrafish have homologous opioidergic genes to humans and a diverse behavioral profile that can be used to validate the neurotoxicity generated by opioids (*Stevens, 2009*; *Demin et al., 2018*). For instance, chronic morphine treatment at 1.5 mg/L induced anxiety-like effects in adult zebrafish (*Cachat et al., 2011*), resulting in a significant increase in time spent near the water surface and a decrease in systemic cortisol levels. Diacetylmorphine (heroin) exposure induced psychostimulant-like hypermobility in adult zebrafish, but did not alter anxiety-like behavior (*Stewart et al., 2015*). The zebrafish conditioned positional preference test is an important behavioral experiment to assess drug reward effects. Adult zebrafish can develop addictive behavior when given substances like fentanyl, oxycodone, bupivacaine, phencyclidine, and chlorpheniramine (*Brock et al., 2017*). Since the opioid system is active during zebrafish embryonic development (*Sarmah et al., 2020*), the zebrafish larvae can be used to examine the toxicity of prenatal opioid exposure. Zebrafish absorb drugs through the embryonic chorion or skin during early stages of embryonic development (*Martins et al., 2018*), whereas adult humans absorb fentanyl through the skin, intravenously, or orally. Despite differences in the mode of ingestion of anesthetics, the teratogenicity of fentanyl has been demonstrated in studies with zebrafish embryos tested. For example, zebrafish embryos have revealed that developmental fentanyl exposure produced cardiotoxicity and respiratory depression, along with reduced swimming ability in larvae (*Kirla et al., 2021*; *Zaig, daSilveira Scarpellini & Montandon, 2021*; *Cooman et al., 2022*). However, these studies have been limited to tests on larvae that have developed to day 5, and only superior behavioral testing on more neurodevelopmental zebrafish larvae can provide a comprehensive evaluation of the neurobehavioral toxicity induced by fentanyl exposure.

Fentanyl and its analogs are ultrashort-acting opioids with selective $\mu$-receptor affinity and gamma aminobutyric acid (GABA) agonist actions, which results in a complex neurotoxic mechanism (*Kim et al., 2017*). Remifentanil, which is a synthetic opioid analgesic drug, binds to $\mu$-opioid receptors and indirectly enhances glutamate toxicity

in patients, resulting in neuronal degeneration and epilepsy (*Angst et al., 2003*). Fentanyl binding to G-protein-coupled receptors also induced excessive dopamine release, which caused oxidative stress and neuronal damage (*Cunha-Oliveira, Rego & Oliveira, 2008*). As with other kinds of nerve damage, disturbance of neurotransmitter homeostasis appears to be involved in the etiology of opioid neurotoxicity (*Kofke et al., 2000*). More research is needed to find out if fentanyl is neurotoxic to developing organisms, especially to see how it affects behavior in animal models.

Fentanyl has a maximum plasma concentration of ug/L levels when used as an analgesic or anesthetic. For example, when fentanyl is used as an anesthetic for corrective surgery for congenital heart defects, the maximum plasma concentration of fentanyl in the patient is $30 \pm 8$ μg/L (*Newland et al., 1989*). We tested fentanyl-induced neurotoxicity in zebrafish larvae at concentrations of 0.1, 1 and 5 mg/L, a range that correlates with blood concentrations of fentanyl following illicit use of fentanyl analogs in humans ($0.00013-2.1$ mg/L) (*Dwyer et al., 2017*). To determine if exposure to fentanyl during the first five days post fertilization causes behavioral changes and neurotoxicity in zebrafish larvae, we evaluated the larvae's performance on days 10–14 post fertilization in terms of photomotor response, anxiety levels, shoaling behavior, aggressive behavior, and behavioral sensitization. The expression of neurological damage marker genes was further examined to provide evidence of neurotoxicity.

## MATERIALS & METHODS

### Chemicals

Fentanyl hydrochloride powder (>99.8% purity, CAS 1443-54-5) was purchased from Shanghai Yuansi Standard Science and Technology Co., Ltd. All other reagents and chemicals were purchased from Sigma-Aldrich (analytical grade, Shanghai). Fentanyl hydrochloride powder was dissolved in distilled water and then diluted to 0.1, 1, 5 mg/L in E3 medium (5 mM NaCl, 0.17 KCl, 0.33 mM $CaCl_2$, and 0.33 mM $MgSO_4$). The Trizol reagent was acquired from Jiancheng (Nanjing, China). Reverse transcriptase kits and SYBR green reagents were purchased from Takara (Dalian, China).

### Zebrafish maintenance and reproduction

In this study, 4–6-month-old zebrafish of the wild type were obtained from the Chinese Zebrafish Resource Center in Wuhan. Prior to the experiment, zebrafish were kept in a 40-liter tank for 15 days at a density of 10 per liter and fed twice daily with brine shrimp. The water in the aquarium system was constantly filtered and oxygenated (pH $6.8-7.2$, $28 \pm 1$ °C, conductivity at 500–600 μS), and the room was lit for 14 h and dark for 10 h (lights on at 8:00 am). Fertilized eggs were collected by generating spawning activity in adult zebrafish. The night before the experiment, male and female zebrafish were positioned in a tank in a 2:1 ratio, separated by a transparent acrylic panel. The next morning at 8:00 am, the divider was removed to trigger natural spawning under the stimulation of light, and eggs were collected within 1 h. Normally developed embryos were chosen for tests following standard techniques (*Howe et al., 2013*). The larvae were kept in an incubator at 28 °C and checked daily for mortality or deformities. Zebrafish larvae do not require additional

feeding for 5 days after fertilization. From 5 to 10 days after fertilization, zebrafish larvae were subjected to a daily straw worm feeding in the morning. From 10 to 14 days after fertilization, zebrafish larvae were fed once daily with fungus shrimp larvae in the morning. At the end of all experiments, zebrafish larvae were anesthetized by ice water and euthanized with 1% sodium hypochlorite. All procedures were approved by the Zhejiang University Experimental Animal Welfare and Ethical Review Committee (Ethical approval number: ZJU20220147).

## Zebrafish exposure to fentanyl

Randomly selected zebrafish embryos were transferred to fentanyl solutions or E3 medium (5 mM NaCl, 0.17 mM KCl, 0.33 mM CaCl$_2$, and 0.33 mM MgSO$_4$) and placed in standard 90 mm Petri dishes within 2 h of fertilization (50 embryos per Petri dish, 146 embryos were usedper concentration). Petri dishes were kept in the incubator at a 14-hour light: 10-hour dark cycle, with 15 mL of exposure solution replaced daily (*i.e.* half of the exposure solution was renewed). After the 2 hpf to 5 days post-fertilization (dpf) exposure test, all zebrafish larvae in the fentanyl and control groups were transferred to E3 solution to begin a five-day recovery period and fed once daily until 14 dpf. Larvae that completed the behavioral tests (Fig. 1) were returned to the petri dishes in the incubator. Different zebrafish were used for different behavioral tests. The mortality rate was less than 5% during the experiment. The absence of significant malformations during zebrafish embryonic development, such as pericardial edema, spinal curvature and delayed yolk sac uptake, was confirmed by stereomicroscopy (Olympus, SZX2). Gene expression associated with neuronal damage was examined in zebrafish larvae at 5 dpf after exposure to fentanyl (Fig. 1).

## Photomotor response

Following methods previously reported (*Tao et al., 2020*), we examined the photomotor response of zebrafish larvae at the end of fentanyl exposure (5 dpf) and at the end of the recovery period (10 dpf). Briefly, 24 larvae per concentration were transferred to 24-well plates containing 1.5 mL of E3 medium per well. The larvae's behavior was monitored utilizing the DanioVision$^{TM}$ observation system (Noldus). Once the zebrafish larvae had acclimated to the dark for 10 min, their responses to three successive 10-minute light at an intensity of 100 lux and 10-minute dark periods were studied. The EthoVision XT 10 software (Noldus) was used to analyze the video data. Anomalous data due to video tracking failures were removed. The swimming distances of larvae during light (visible light) and dark (infrared light) periods were examined.

## Thigmotaxis behavior

Zebrafish prefer to swim to the edge, which is associated with anxiety (*Yang et al., 2017*). At 10 dpf, a total of 24 randomly selected larvae per concentration were transferred to 24-well plates according to the previous report (*Yang et al., 2017*). After an acclimation period of 30 min, the behavioral characteristics of zebrafish were then recorded under 10 min of light (intensity of 100 lux) and 10 min of darkness for a total of 20 min. In the 24-well plate, a circle was created in the middle of each well, with the inner circle's area equal to half of

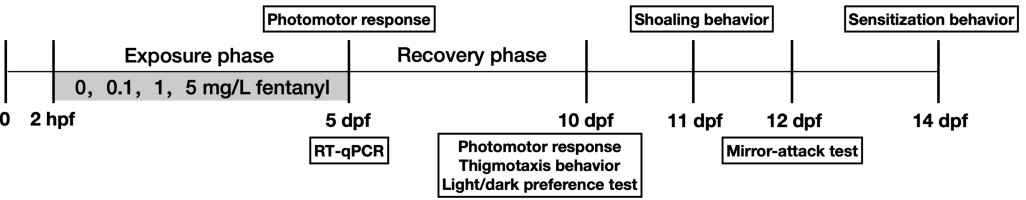

**Figure 1** A schematic diagram showing zebrafish embryos exposed to 0, 0.1, 1, 5 mg/L fentanyl and sampled at different time points for behavior tests.

each well's total area. The EthoVision XT 10 software was used to measure and analyze the time in the central zone and entries into central zone as two thigmotaxis parameters.

## Light/dark preference test

24 larvae were taken at random from each concentration group at 10 dpf and placed into the wells of the 24-well plate (1 mL of E3 medium in each well). The bottom of the 24-well plate was covered with a black and white acrylic plate, which divided the circular wells into two large equal areas of light and dark zones (*Chen et al., 2018*). The DanioVision™ system was used to collect data on zebrafish behavior under light conditions every 60 s for 6 min. The amount of time zebrafish larvae spent in the light zone and the number of times they moved from the light zone to the dark zone were used to examine zebrafish preference for light and dark.

## Shoaling behavior

Following methods previously reported (*Chen et al., 2018*), we examined the shallow behavior of zebrafish larvae at 11 dpf. In brief, 20 larvae were picked at random from each concentration group and placed in 90-mm glass Petri dishes (10 larvae and 30 mL of E3 medium per Petri dish). The movements of zebrafish were tracked for 8 min, with data being collected every 15 s. The first 2 min were used to get the fish used to visible light, and the last 6 min were used to study how the fish behaved when they were in groups. Zebralab software was used to calculate the inter-individual distance (IID) and the nearest neighbor distance (NND) as indicators of shoaling behavior.

## Mirror-attack test

At 12 dpf, 12 larvae were taken at random from each fentanyl concentration group and transferred to rectangular 6-chamber polypropylene dishes (North Star Plastics in Jinan District, China) for mirror-attack studies. A mirror paper (3.5 × 1.5 cm) was placed to one side of each chamber (3.5 × 3.5 × 1.5 cm) of the dish (*Chen et al., 2021*). Zebrafish in separate chambers were unable to see one another. The behavior of zebrafish was monitored using a ZebraLab behavioral monitoring station for 8 min under light conditions, with data collected every 60 s. The attack zone was established at 1 cm from the mirror paper using EthoVision XT 10 software. The number of times the fish entered the attack zone and the amount of time zebrafish spent in the attack zone were determined.

## Sensitization behavior

At 14 dpf, 12 zebrafish larvae from each concentration group were randomly moved to a 0.01 mg/L fentanyl solution with 10 min _s_ and quickly transferred to 96-well plates (300 µL of 0.01 mg/L fentanyl solution per well). The behavior of the larvae was monitored. Once the zebrafish larvae had acclimated to the dark for 10 min, their responses to three successive 10-minute light (a light intensity of 100 lux) and 10-minute dark periods were studied. The swimming distances of larvae during light (visible light) and dark (infrared light) periods were examined to determine the behavioral effects of repeated exposure of larvae to the fentanyl.

## Quantitative real-time polymerase chain reaction (qPCR)

Changes in transcript levels regulating neural activity (_bdnf_, _c-fos_) as well as neuronal development and plasticity (_npas4a, egr1, btg2, ier2a, vgf_) were examined in zebrafish larvae at 5 dpf after fentanyl exposure. Thirty larvae were randomly selected from each concentration group and total RNA was extracted using Trizol reagent (Takara, Osaka, Japan) after hypothermia anesthesia.RNA was quantified by measuring the absorbance at 260 nm and diluted to 200 ng/L. Takara's cDNA first strand synthesis kit (Dalian, China) was then used to reverse-transcribe 1 µg of total RNA. Following the manufacturer's protocol, quantitative analysis by qPCR was performed using the CFX-96™ Real-Time PCR Detection System (BioRad, USA). The cycling conditions were as follows: 95 °C for 3 min, followed by 40 cycles of 95 °C for 30 s, 60 °C for 30 s, and 72 °C for 30 s. $\beta$-actin expression levels were used to normalize the relative expression ratios of neural-related genes, which were then evaluated using the $2^{-(\Delta\Delta Ct)}$ method (_Livak & Schmittgen, 2001_). The primer sequences are provided in Table S1.

## Statistical analysis

Normal data were described by mean and standard error of the mean, while non-parametric data were described by median and interquartile range. One-way analysis of variance (ANOVA) was used to investigate differences between the treatment and control groups. The normality of the behavioral data was tested using Kolmogorov-Smirnov, and the homogeneity of variance was tested using Bartlett's test. Welch's ANOVA was used when the data did not meet the assumption of homogeneity of variance. Each treatment was compared to the control group using the appropriate post hoc test (Dunnett's T3 multiple comparison test) when differences were found to be statistically significant. Prism 9 (GraphPad, USA) was utilized to analyze significant differences, and statistical significance thresholds for all experiments were set at $^*p < 0.05$, $^{**}p < 0.01$, $^{***}p < 0.001$ and $^{****}p < 0.0001$.

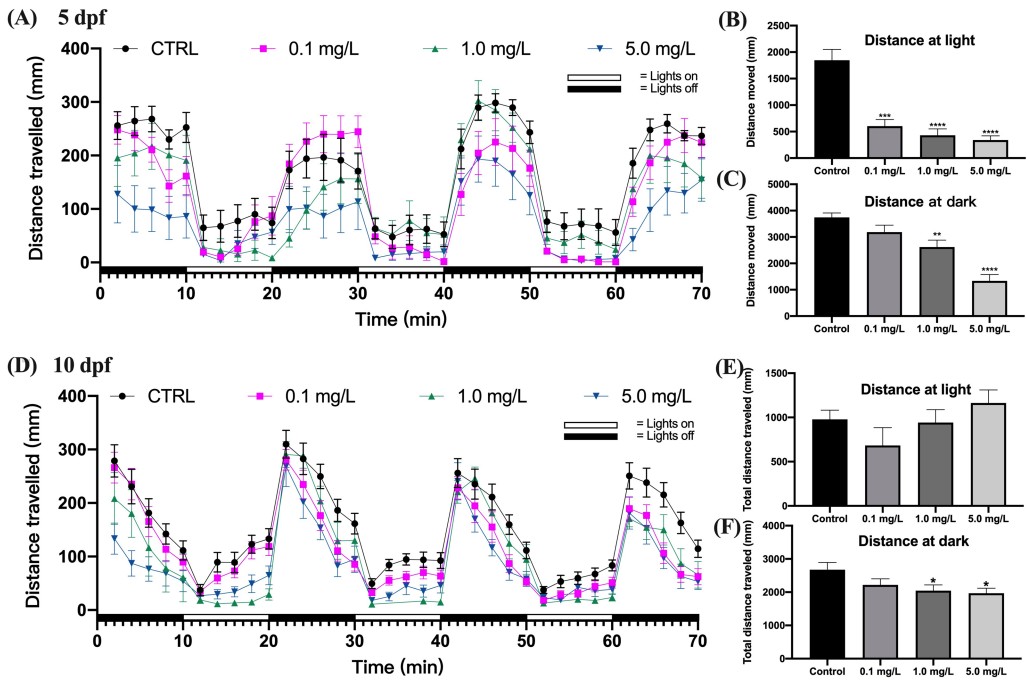

**Figure 2 Effect of fentanyl exposure on the locomotor activity of zebrafish larvae at 5 dpf and 10 dpf.** Behavioral tests were performed on zebrafish larvae after they were exposed to 0.1, 1, and 5 mg/L fentanyl in E3 medium from 2 h post fertilization to 5 dpf and 10 dpf ($n = 3$). (A) Distance traveled following fentanyl exposure at 5 dpf. (B) Total distance travelled during the light period at 5 dpf. (C) Total distance travelled during the dark period at 5 dpf. (D) Distance traveled after the recovery period at 10 dpf. (E) Total distance travelled during the light period at 10 dpf. (F) Total distance travelled during the dark period at 10 dpf. Values are plotted as mean $\pm$ SEM. Significance was defined as *$p < 0.01$, **$p < 0.01$, ***$p < 0.001$, ****$p < 0.0001$.

## RESULTS

### Fentanyl exposure resulted in decreased behavioral capacity

The behavioral capacity of zebrafish larvae at 5 dpf was evaluated by the distance traveled under light and dark stimuli. The findings indicated that zebrafish larvae moved farther during the dark period than they did during the light period (Fig. 2A). Fentanyl exposure led to a concentration-dependent decrease in the average total moving distance of zebrafish larvae under both light and dark conditions (Figs. 2B and 2C). When compared to the control group, zebrafish larvae exposed to 0.1, 1, or 5 mg/L of fentanyl experienced significant reductions in the amount of distance they traveled during the light phase (Fig. 2B) (Welch's ANOVA test, $F (3.000, 45.06) = 15.61$, $p < 0.0001$; Dunnett's T3 post hoc test, all $p < 0.0001$) . Only the group that was given 0.1 mg/L of fentanyl did not experience a significant reduction in travel distance during the dark phase in comparison to the control larvae (Fig. 2C) (one-way ANOVA, $F (3, 87) = 17.58$, $p < 0.0001$; Dunnett's T3 post hoc test, $p = 0.2158$ for 0.1 mg/L, $p = 0.0024$ for 1 mg/L, $p < 0.0001$ for 5 mg/L).

Fentanyl continued to impair the behavioral abilities of zebrafish larvae after a 5-day recovery interval (Fig. 2D). Although there were no significant differences in the travel
distance in the control and fentanyl-exposed groups during light periods (Fig. 2E) (one-way ANOVA, F (3, 81) = 1.666, $p$ = 0.1808), zebrafish that had been exposed to 1 or 5 mg/L fentanyl up to 5 dpf and then recovered for 5 days, traveled significantly smaller distances than control fish in the dark conditions (Fig. 2F) (one-way ANOVA, F (3, 85) = 3.096, $p$ = 0.0311; Dunnett's T3 post hoc test, $p$ = 0.0429 for 1 mg/L, $p$ = 0.0199 for 5 mg/L).

**Fentanyl exposure resulted in increased anxious behavior**

When exposed to fentanyl for 5 dpf and then recovered for 5 days in the clean medium, zebrafish larvae showed a decrease in their preference for the central region (thigmotaxis), which indicated increased anxiety. Compared to control fish, 10 dpf larvae exposed to fentanyl spent significantly smaller time in the central region (Fig. 3A) (Welch's ANOVA test, F (3.000, 42.71) = 5.937, $p$ = 0.0018; Dunnett's T3 post hoc test, $p$ = 0.0135 for 0.1 mg/L, p = 0.0069 for 1 mg/L, $p$ = 0.0004 for 5 mg/L) and made significantly fewer entrances into the central region (Fig. 3B) (one-way ANOVA, F (3, 85) = 11.66, $p$ < 0.0001; Dunnett's T3 post hoc test, $p$ < 0.0001 for 0.1 mg/L, $p$ = 0.0003 for 1 mg/L, $p$ = 0.0011 for 5 mg/L). Another important indicator of high levels of anxiety in zebrafish larvae is the increased preference for light (*Bai et al., 2016*). Here, the time zebrafish spent in the light region was positively correlated with fentanyl concentration (Fig. 3C), and all concentrations were significantly increased compared to the control (Welch's ANOVA test, F (3.000, 42.79) = 35.42, $p$ < 0.0001; Dunnett's T3 post hoc test, $p$ = 0.0347 for 0.1 mg/L, $p$ < 0.0001 for 1 mg/L, $p$ < 0.0001 for 5 mg/L). Also, compared to the control group, the larvae in the 1 mg/L and 5 mg/L fentanyl groups moved between the light and dark areas significantly more often (Fig. 3D) (one-way ANOVA, F (3, 89) = 4.537, $p$ = 0.0052; Dunnett's T3 post hoc test, $p$ = 0.0057 for 1 mg/L, $p$ = 0.0064 for 5 mg/L).

**Fentanyl exposure produced deficient social behavior**

For shoaling at 11 dpf, zebrafish exposed to 1 or 5 mg/L fentanyl showed significantly increased inter-individual distance (IID) (one-way ANOVA, F (3, 188) = 8.934, $p$ < 0.0001; Dunnett's T3 post hoc test, $p$ = 0.0003 for 1 mg/L, $p$ = 0.0011 for 5 mg/L) and increased nearest neighbor distance (NND) (Welch's ANOVA test, F (3.000, 65.59) = 5.363, $p$ = 0.0023; Dunnett's T3 post hoc test, $p$ = 0.0038 for 1 mg/L, $p$ = 0.0023 for 5 mg/L) compared to control fish (Figs. 4A and Fig. 4B). Zebrafish larvae exposed to fentanyl at 1 or 5 mg/L entered the mirror zone significantly less frequently than control fish (Fig. 4C) (one-way ANOVA, F (3, 44) = 6.621, $p$ = 0.0009; Dunnett's T3 post hoc test, $p$ = 0.0182 for 1 mg/L, $p$ = 0.0004 for 5 mg/L). Although the fentanyl-exposed fish spent more time in the mirror region than control fish, the differences were not significant (Fig. 4D) (one-way ANOVA, F (3, 44) = 0.8351, $p$ = 0.4818).

**Behavioral sensitization to a fentanyl challenge at 0.01 mg/L**

Behavioral sensitization refers to the enhanced motor stimulus response that occurs during repetitive, intermittent exposure to a specific drug, and in the present study we evaluated this by the motor ability of zebrafish in photomotor response. The locomotor response to a repeated fentanyl stimulus at 14 dpf was studied in zebrafish that had been exposed to fentanyl at 0.1, 1 or 5 mg/L during the first 5 dpf, then recovered in clean E3 medium up to

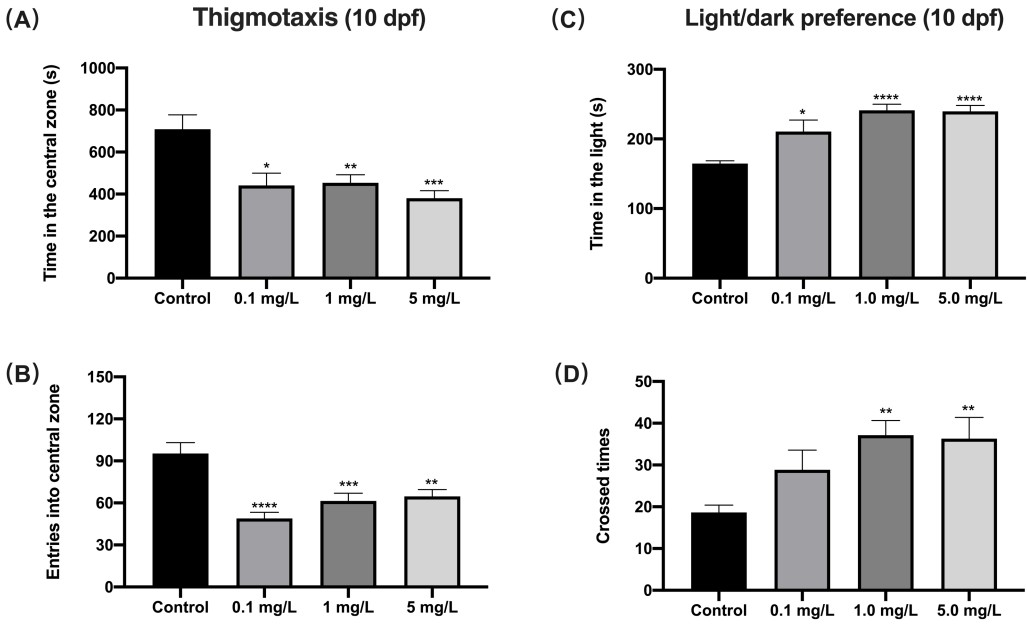

**Figure 3** **The effect of fentanyl exposure on larval thigmotaxis behavior and light/dark preference.** Zebrafish embryos were exposed to 0.1, 1, and 5 mg/L fentanyl in E3 medium from 2 h post fertilization to 5 dpf, followed by a 5-day recovery interval, and behavioral tests were performed at 10–11 dpf. (A–B) Light/dark background preference tests were performed on larvae at 10 dpf ($n = 3$). (C–D) Thigmotaxis behavior was assessed on larvae at 11 dpf ($n = 3$). Values are plotted as mean ± SEM . Significance was defined as $^*p < 0.05$, $^{**}p < 0.01$, $^{***}p < 0.001$, $^{****}p < 0.0001$.

14 dpf. Zebrafish larvae exposed to fentanyl during development experienced behavioral changes after re-exposure to fentanyl (Fig. 5). After the 0.01 mg/L fentanyl challenge, zebrafish larvae in the 5 mg/L fentanyl group showed significant increases in swimming distance during light (Fig. 5B) (one-way ANOVA, $F (3, 37) = 3.473$, $p = 0.0255$; Dunnett's T3 post hoc test, $p = 0.0297$) and dark periods (Fig. 5C) (one-way ANOVA, $F (3, 37) = 3.864$, $p = 0.0168$; Dunnett's T3 post hoc test, $p = 0.0080$).

## Fentanyl exposure alters the expression of genes related to neuro-development

We further examined how different concentrations of fentanyl affected the transcriptional regulation of genes regulating neural activity (*bdnf, c-fos*) and neuronal development and plasticity (*npas4a, egr1, btg2, ier2a, vgf*). At 5 dpf, fentanyl exposure at a concentration of 5 mg/L significantly increased the relative expression of *bdnf* (one-way ANOVA, $F (3, 8) = 9.878$, $p = 0.0046$; Dunnett's T3 post hoc test, $p = 0.0181$), *c-fos* (one-way ANOVA, $F (3, 8) = 5.327$, $p = 0.0261$; Dunnett's T3 post hoc test, $p = 0.0172$), *napas4a* (one-way ANOVA, $F (3, 8) = 23.49$, $p = 0.0003$; Dunnett's T3 post hoc test, $p = 0.0003$), *egr1* (one-way ANOVA, $F (3, 8) = 12.199$, $p = 0.0024$; Dunnett's T3 post hoc test, $p = 0.0090$) in comparison to control larvae (Figs. 6A–6D). At concentrations 1 and 5 mg/L, fentanyl exposure significantly upregulated *btg2* (one-way ANOVA, $F (3, 8) = 76.72$, $p < 0.0001$; Dunnett's T3 post hoc test, $p = 0.000$ for 1 mg/L, $p < 0.0001$ for 5 mg/L) and *ier2a* (one-way

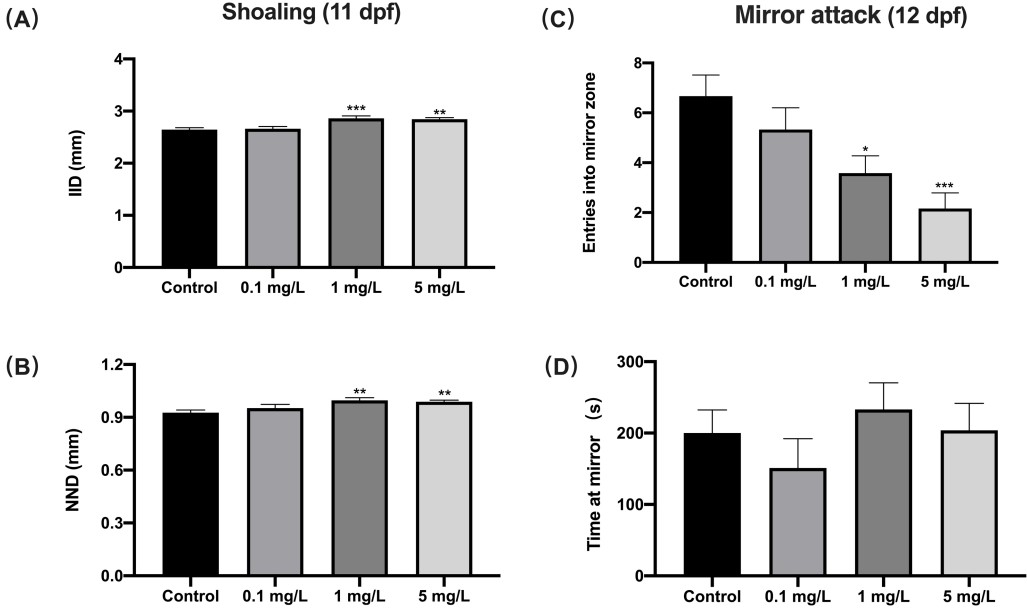

**Figure 4** **The effect of fentanyl exposure on shoaling and mirror attack behavior of larvae.** Zebrafish embryos were exposed to 0.1, 1, and 5 mg/L fentanyl in E3 medium from 2 h post fertilization to 5 dpf, followed by a 5-day recovery interval, and behavioral tests were performed 11–12 dpf. (A–B) Shoaling behavior evaluated using larvae at 11 dpf ($n = 3$). Inter-individual distance (IID) and nearest neighbor distance (NND). (C–D) Mirror attack behavior evaluated using larvae at 12 dpf ($n = 3$). Values are plotted as mean ± SEM. Significance was defined as $^*p < 0.05$, $^{**}p < 0.01$, $^{***}p < 0.001$.

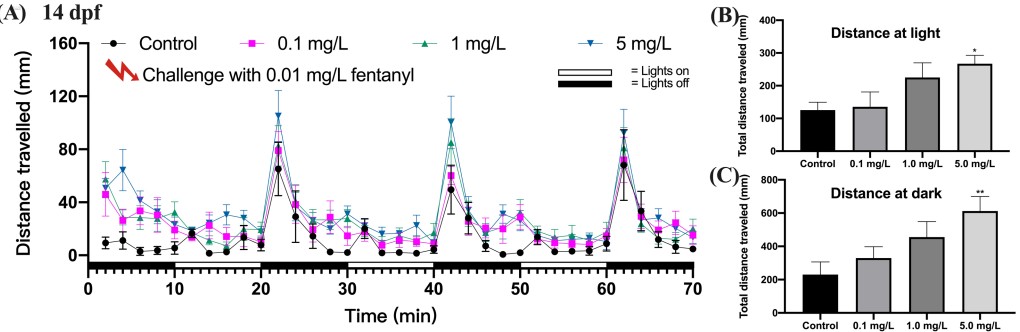

**Figure 5** **Behavioral sensitization after re-exposure to 0.01 mg/L fentanyl at 14 dpf.** Zebrafish embryos were exposed to 0.1, 1, and 5 mg/L fentanyl in E3 medium from 2 h post fertilization to 5 dpf, followed by a 5-day recovery interval, and behavioral tests were performed at 14 dpf (n = 3). (A) Distance travelled every 2 min after the fentanyl challenge. (B) Total distance travelled during the light period. (C) Total distance travelled during the dark period. Values are plotted as mean ± SEM. Significance was defined as $^*p < 0.05$, $^{**}p < 0.01$.

ANOVA, F (3, 8) = 11.51, $p = 0.0028$; Dunnett's T3 post hoc test, $p = 0.0116$ for 1 mg/L, $p = 0.0013$ for 5 mg/L) genes relative to controls (Figs. 6E and 6F). For *vgf*, all the fentanyl groups showed significant increase relative to control group (Fig. 6G) (one-way ANOVA,

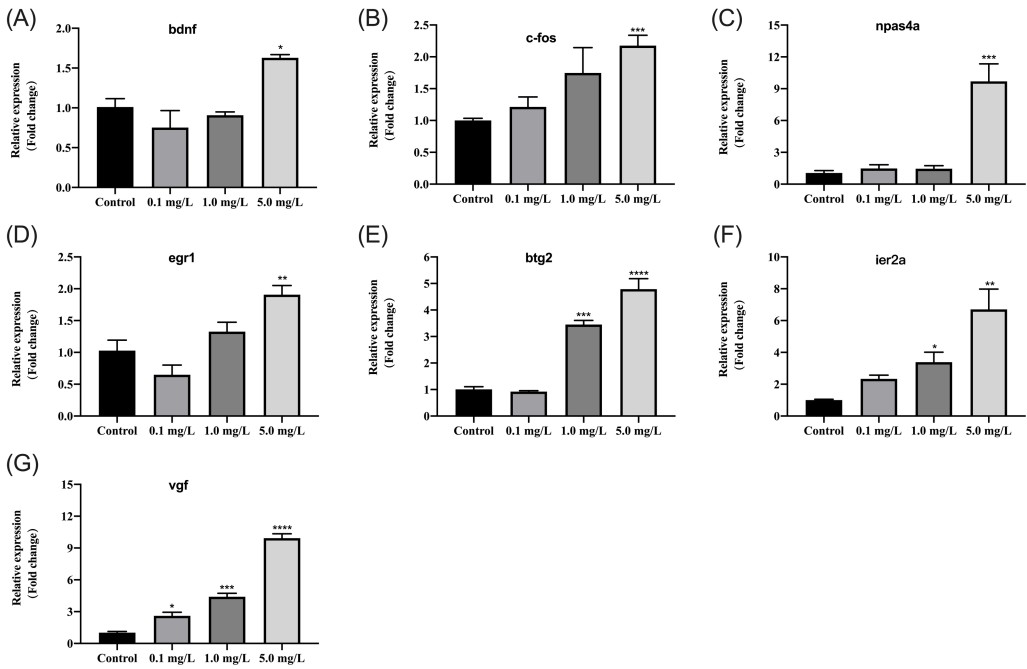

**Figure 6  Relative mRNA expression levels of neurodevelopmental marker genes in zebrafish at 5 dpf after fentanyl exposure.** (A–B) Genes regulating neural activity (*bdnf*, *c-fos*). (C–G) Genes regulating neuronal development and plasticity (*egr1*, *napas4a*, *vgf*, *btg2* and *ier2a*). Data are expressed as mean ± SEM. Significance was defined as $^*p < 0.05$, $^{**}p < 0.01$ and $^{***}p < 0.001$.

$F (3, 8) = 152.4$, $p < 0.0001$; Dunnett's T3 post hoc test, $p = 0.0184$ for 0.1 mg/L, $p = 0.000$ for 1 mg/L, $p < 0.0001$ for 5 mg/L).

## DISCUSSION

In recent years, zebrafish have been widely utilized as a model aquatic vertebrate for toxicological research. Thus far, research on the toxicological effects of fentanyl on zebrafish has focused on cardiac development, respiratory depression, and behavioral inhibition. For example, in the zebrafish embryotoxicity (zFET) test, the lethal concentration of fentanyl was 83.17 μM (28.40 mg/L) and the sublethal value (endpoint including pericardial edema, irregular heartbeat, yolk sac edema, and head cord deformities) was 16.15 μM (5.52 mg/L) in 5 dpf zebrafish larvae (*Kirla et al., 2021*; *Cooman et al., 2022*). The range of fentanyl concentrations (0.1–5 mg/L) used in the current study exhibited no teratogenic effects on zebrafish development, but caused behavioral alterations in numerous motor and social behavioral tests, demonstrating that low concentrations of fentanyl could cause neurobehavioral toxicity.

Neurodevelopment in zebrafish embryos is a complex process and if, at any stage, the process is altered, interrupted or inhibited, the consequences could be permanent. We chose to expose zebrafish embryos to fentanyl at 4 hpf to 5 dpf, this was in consideration of the fact that the major organs of the zebrafish would have completed their developmental maturation during this period. A growing number of studies have used various behavioral

endpoints in zebrafish to assess the effects of chemicals and contaminants (*Rosa, Lima & Lopes-Ferreira, 2022*; *Zhang et al., 2021*). As soon as possible after undergoing the recovery period, we tested for complex behaviors, which are usually not reliably measured at 5 dpf. Complex behaviors (for example, social behavior, memory, and learning) can reveal more about toxicological effects than simple behavioral analyses of model animals (*Hong & Zha, 2019*). Our findings demonstrate that exposure to fentanyl during embryo development has a concentration-dependent effect on anxiety, shoaling, aggression, social preference, and sensitization in zebrafish. Zebrafish larvae show a stable motor response to continuous alternating light and dark stimuli at 6 dpf, *i.e.* light inhibits behavior and dark promotes behavior, which needs supraspinal input from the brain and is a crucial sign for determining neurodevelopment (*Hong & Zha, 2019*). In our investigation, fentanyl administration reduced the distance traveled of 5 dpf larvae during cycles of light and dark. Other opioids revealed similar toxicity in zebrafish larvae. For example, zebrafish displayed decreased swimming speed after exposure to different fentanyl analogues at 1 μM (0.34 mg/L for fentanly and 0.35 mg/L for butyrfentanyl) (*Pesavento et al., 2022*). The high sensitivity of fentanyl for mammalian μ-type opioid receptors and the similarities between zebrafish and mammalian opioid systems (*Steenbergen, Richardson & Champagne, 2011*) make zebrafish useful for studying fentanyl toxicity in animals and even humans. *Oprm1* (Opioid receptor mu1) expression is detectable in numerous nerve centers at 24 h post fertilization in zebrafish embryos (*Sarmah et al., 2020*). The motor impairment caused by fentanyl in zebrafish may be a result of the analgesic and sedative effects provided by the binding of opioid receptors (*Gampfer et al., 2020*; *Zaig, daSilveira Scarpellini & Montandon, 2021*). In fact, after the recovery period, we observed no significant difference in the behavioral capacity of zebrafish larvae exposed to all concentrations of fentanyl under bright conditions compared to the control group, indicating that behavioral capacity under light conditions was recovered. However, the behavioral abilities of zebrafish larvae exposed to 1.0 mg/L and 5.0 mg/L concentrations of fentanyl under dark conditions remained significantly lower than those of the control group, which may be due to the anxiety-induced effects on larval behavior under non-preferred conditions (darkness) (*Gampfer et al., 2020*).

During the later phases of zebrafish larval development after the recovery period, anxiety, sociability, and other behaviors were assessed further. Commonly, zebrafish thigmotaxis tests and light/dark preference tests have been employed to evaluate the anxiety-inducing effects of substances (*Peng et al., 2016*; *Yang et al., 2017*; *Chen et al., 2018*). Zebrafish larvae like to swim near the edge and have a strong dark avoidance behavior (*Steenbergen, Richardson & Champagne, 2011*). In our investigation, fentanyl dramatically decreased the proportion of time spent in light areas while increasing the propensity to explore intermediate locations, indicating that fentanyl administration enhanced zebrafish anxiety. Opioid-like anxiety behaviors have been found in rats on long-term morphine use and have been demonstrated to include functional abnormalities in the limbic dopaminergic system (*Raghav et al., 2021*; *Deji et al., 2022*).

Zebrafish are highly social animals that prefer swimming in groups (*Mansur Bde et al., 2014*). The increase of IID and NND in the shoaling test indicated that zebrafish have decreased social cohesion (*Chen et al., 2018*). Typically, zebrafish do not begin to shoal

until a few days after hatching, and this propensity steadily develops during development. Zebrafish undergo alterations in their dopaminergic and serotonergic systems as their shoal behavior matures. Disruption of early neurodevelopment (*Chen et al., 2021*; *Fu et al., 2021*; *Wang et al., 2022*) and neurotransmitter systems (*Santos et al., 2021*) in the zebrafish brain may be responsible for reduced shoal behavior. Another well-established tool for researching zebrafish sociality is the mirror attack test, which is used to assess zebrafish aggression (*Qiu et al., 2021*). We found that the exposure to fentanyl enhanced the frequency of mirror attacks and the proportion of time spent in the attack area. Even though there haven't been many behavioral studies of opioid withdrawal in fish, chronic aggression has been observed in mouse models after opioid withdrawal (*Piccin & Contarino, 2020*), and morphine made both male and female mice less social (*Piccin, Courtand & Contarino, 2022*).

Sensitization to addictive drugs plays a significant role in the onset and recurrence of relapse and drug-seeking behavior. In this study, embryonic fentanyl exposure in zebrafish resulted in an enhanced behavioral response to the fentanyl challenge. Zebrafish exhibit stronger behavioral responses after repeated exposure to cocaine, nicotine, and ethanol, which is consistent with our findings (*Pisera-Fuster et al., 2019*). In adult zebrafish, behavioral sensitization generated by a single morphine exposure resulted in significant behavioral sensitization after repeated and intermittent morphine (1.25–10 µM, 0.36−2.85 mg/L) treatment (*Bian et al., 2022*). The dosages employed to elicit behavioral sensitization in zebrafish are typically low, owing to the fact that excessive concentrations produce a decrease in locomotor activity. The behavioral sensitization we found with fentanyl in zebrafish may be due to a negative reinforcement mechanism, *i.e.*, a heightened behavioral response generated by ceasing or avoiding an aversive stimulus such as a withdrawal medication (*Koob, 2013*), which should be further validated on other animal models.

The genes addressed in this article that were related to essential neuronal activities may be early markers of potentially harmful impacts on neural development in zebrafish embryos, which can result in altered organismal behavior, memory, and even energy balance. Brain-derived neurotrophic factors (*bdnf*) play critical roles in neuronal cell proliferation, differentiation, and maintenance, as well as synaptic plasticity, learning, and long-term memory. The neurotrophic factor bdnf has been identified as a possible downstream target of c-fos and *npas4a* (*Lin et al., 2008*; *Lu, 2011*). In zebrafish, a reduction in aggressive behavior was observed to be mediated by an increase in *bdnf* expression in the brains (*Theodoridi, Tsalafouta & Pavlidis, 2017*). The *c-fos* is an excellent biomarker for identifying synaptic activation (*Torres-Hernandez et al., 2016*), and the relative expression of *c-fos* was more than twice as high in zebrafish exposed to high fentanyl as it was in the control group. Experiments on mice revealed a correlation between *c-fos* expression in a specific subset of brain areas and opioid hyperalgesia and physical withdrawal symptoms (*Alvarez-Bagnarol et al., 2022*). In addition, the expression level of *c-fos* mRNA in the zebrafish brain increased considerably following nicotine abstinence in the zebrafish nicotine withdrawal experiment (*Ponzoni et al., 2021*).

The *npas4a* homolog is a brain-specific activity dependent gene involved in forebrain development. For example, *npas4a* gene deletion during embryonic development resulted

in forebrain defects and reduced expression of specific forebrain markers (*Lin et al., 2008*). As part of the control of excitation-inhibition equilibrium in brain circuits, *npas4a* has also been shown to be upregulated in proportion to the severity of seizures (*Bloodgood et al., 2013*). Animals' learning and behavior in regards to fear have recently been linked to changed amounts of *npas4a* expression in zebrafish (*Torres-Hernandez et al., 2016*; *Baker & Wong, 2021*). The *egr1* is an inducible transcription factor in the brain that regulates synaptic plasticity, memory, as well as neuronal cell development and differentiation (*Boyer, Ernest & Rosa, 2013*). It has been proposed that b-cell translocation gene 2 (*btg2*) and *ier2a* are key regulators of vertebrate neural development (*Moriya et al., 2016*). Previous investigations associating *btg2* deletion in zebrafish and mice with aberrant aggression raise the possibility that both genes are crucial for brain processes related to aggression (*Malki et al., 2016*; *Liu et al., 2020*). The *vgf* (VGF nerve growth factor-inducible), a neuropeptide involved in crucial neural processes such as synaptic plasticity, neurogenesis, and brain neurite growth, is produced by brain-derived neurotrophic and nerve growth factors (*Mizoguchi et al., 2017*; *Takeuchi et al., 2018*) and has been shown to have neuroprotective effects on retinal ganglion cells (*Takeuchi et al., 2018*). In addition, *vgf* knockout led to increased sadness and antisocial behavior in mice, as well as decreased brain weight and memory impairment (*Mizoguchi et al., 2017*). Depression, schizophrenia, and bipolar disorder are related to aberrant levels of *vgf* in the human brain, indicating that this gene is involved in the pathophysiology of mental diseases (*Lin et al., 2015*).

## CONCLUSIONS

In conclusion, we reveal for the first time that exposure of zebrafish embryos to fentanyl resulted in elevated anxiety, decreased social preference and aggressiveness, and induced behavioral sensitization. The expression of genes showed that embryonic exposure to fentanyl at concentrations of 5 mg/L caused substantial alterations in neural activity (*bdnf*, *c-fos*) and neuronal development and plasticity (*npas4a*, *egr1*, *btg2*, *ier2a*, *vgf*). The results demonstrated the neurotoxicity of fentanyl at nonteratogenic concentrations to developing organisms, providing evidence for the risk of fentanyl use during pregnancy.

### Funding

This work was supported by the Zhejiang Provincial Natural Science Foundation of China under Grant (LGF20C090001), the Zhejiang Province Key Research and Development Program (No. 2021C03135) and the Open Project of Zhejiang Provincial Key Laboratory of Drug Prevention and Control Technology Research (2020013). The funders had no role in study design, data collection and analysis, decision to publish, or preparation of the manuscript.

### Grant Disclosures

The following grant information was disclosed by the authors:

Zhejiang Provincial Natural Science Foundation of China under Grant: LGF20C090001.
Zhejiang Province Key Research and Development Program: 2021C03135.
Open Project of Zhejiang Provincial Key Laboratory of Drug Prevention.
Control Technology Research: 2020013.

## Competing Interests

The authors declare there are no competing interests.

## Author Contributions

- Binjie Wang conceived and designed the experiments, analyzed the data, authored or reviewed drafts of the article, and approved the final draft.
- Jiale Chen performed the experiments, authored or reviewed drafts of the article, and approved the final draft.
- Zhong Sheng performed the experiments, authored or reviewed drafts of the article, and approved the final draft.
- Wanting Lian performed the experiments, analyzed the data, prepared figures and/or tables, and approved the final draft.
- Yuanzhao Wu analyzed the data, prepared figures and/or tables, and approved the final draft.
- Meng Liu analyzed the data, authored or reviewed drafts of the article, and approved the final draft.

## Animal Ethics

The following information was supplied relating to ethical approvals (i.e., approving body and any reference numbers):

The Zhejiang University Experimental Animal Welfare and Ethical Review Committee approved the study (Ethical approval number: ZJU20220147).

## Data Availability

The raw measurements are available in the Supplementary Files.

## Supplemental Information

Supplemental information for this article can be found online at http://dx.doi.org/10.7717/peerj.14524#supplemental-information.

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
