# Peer review of "Embryonic exposure to fentanyl induces behavioral changes and neurotoxicity in zebrafish larvae"

_PeerJ, doi:10.7717/peerj.14524_

## Round 0.1 · original submission · Major Revisions

· Academic Editor

Major Revisions

Dear authors,

Four reviewers have evaluated the manuscript and provided suggestions for improving the clarity of the manuscript. As you can see, the reviewers are concerned about the statistical validity of the results. Please pay special attention to providing sufficient experimental data and information about statistical analysis performed when responding to the reviewers’ comments. Also, one of the major concerns was the lack of justification for the exposure dosage and the exposure route. Please indicate in your response letter how each of the reviewer comment was incorporated in the manuscript text and if it was not, please explain the reasons in the response letter.

Reviewer 1 ·

Basic reporting

The mansucript is well written supported by recent literature and providing sufficient background for the study. The structure meets the journal standards and raw data is shared.

Experimental design

The experimental design is welld escribed although some issues need to be solved. In particular, teh experimental unit was the petri dish and not the fish. Therefore, auhtors have a n=3 and not n=24 or 20 or other as described in the manuscript. The research question/hypothesis should be included. The methods also need to be complemented by further information in order to this study be replicated.

Validity of the findings

Data is provided and statistical methods sound. Conclusions are well stated and associated to the research objectives. The impact and nvelty should be further described.

Additional comments

The authors present a study on the neurotoxic effects of fentanyl, a powerful opioid used as a pain medication and for anesthesia, using zebrafish larvae as the animal model. Embryos were exposed from 2 hpf to 5 dpf to fentanyl with a further recovery period in which larvae were maintained in E3 medium. At different time-points (5, 10, 12 and 14 dpf), behavioural paradigms were applied to the larvae to assess the neurotoxic effects. In addition, neuronal damage was further complemented with gene expression changes by qPCR.

Although being an interesting study about the neurotoxicological effects of this compound in this aquatic model which has been shown to have a preserved opioid pharmacology similar to higher vertebrates, some issues need to be addressed:

Major comments:

- The introduction should be revised taking in consideration the opiod administration differences between zebrafish and humans and therefore the consequent toxicological effects (the introduction is too focused on human effects). Furthermore, at these early developmental stages, the compound is absorbed by the embryonic chorion although drugs can be absorbed through the skin and via swallowing contrary to the human absorption via.
- While the results of this study are novel, the idea is not new as some studies have already shown behavioural effects of fentanyl in this animal model. Some of these references (e.g. Pesavento 2022) are already included in the discussion. This information should also be given in the introduction and the novelty of this study highlighted rather than describing effects of other opioids.
- Why is the CPA test and the addictive effects in adults described in the introduction?
- The work of Dwyer et al 2017 describe the blood concentrations of a fentanyl analog after overdose death. What are the levels of fentanyl used for analgesia and anesthesia?
- Why were larvae anesthetized by ice water followed by 1% sodium hypochlorite? Ice water is enough to euthanize larvae. What are the effects/consequences of using sodium hypochlorite on the evaluated (gene expression) parameters?
- According to the experimental design, 50 embryos per petri dish were used with 24 larvae used for photomotor response, 24 also used for thigmotaxis, and another set of 24 used for light/dark preference test. Were these the same animals? Were mortalities observed? Were new experiments conducted? Further information should be given for the procedures.
- Why three different behavioural softwares (EthoVision XT10, DanioVision and Zebralab) were used?
- Regarding the results, as the exposure was conducted in each Petri dish, all the 50 embryos should be considered as n=1. As such, when describing the results, authors could only have a total observations of 12 (x animals=1 x 3 replicates x 4 groups).
- Why was anxious behaviour only assessed at 10 dpf and not after the end of the exposure, at 5 dpf?
- How does the gene expression changes determined at 5 dpf relate with the behavioural effects after the recovery period?
- L293, yet significant changes were observed at 5 dpf with light stimulation.
- L303, was the analgesic and sedative effects evaluated to confirm these are effective analgesic concentrations? Was a inhibitor of fentanyl effects used (such as naloxone)?
- All the behavioural discussion should be revised to discuss the findings taking in consideration the recovery period. In fact, the exposure lasted for 5 days and most of the significant results were still found after a 5 days recovery period. Yet, that is not discussed. Is this drug maintained in the larvae after exposure? Are these long-lasting effects due to early disruption of developmental processes? In fact, multiple changes were observed and described for different genes associated to neural development. This should be discussed.
- The conclusion should be revised as I believe the usefulness of zebrafish to study opioid effects has been previously shown.

Minor comments:

- L76, concentrations as well as their rationale should be given in the methods section.
- L90, E3 composition should be described here.
- L121, gene expression analysis is missing in this scheme.
- L122, which teratogenic effects were observed? According to the literature, doi: 10.1002/jat.4253, fentanyl induces teratogenic effects in this species.
- L156, please confirm if these 30 mL fit the 90-mm Petri dishes.
- L173, has this test been previously described?
- L180, the genes should be introduced here as well as their respective function (L261-263 should be moved to this section).
- L186, further information (e.g. temperature program, equipment, etc) is required for the qPCR analysis.
- L191, normal data should be described by mean and SD while non-parametric data should be described by the median and IQR.
- Figure 2 and 3 could be merged.
- L223, the results section should not contain references.
- L285, where are these results shown?
- Gene expression analysis is missing from figure 1.
- As referred before, n should be equal 3 not 24 as described in Figure 4. The same for figure 5.
- Primers should contain the accession number and the specific temperature.

Reviewer 2 ·

Basic reporting

The article was very well structured, with sufficient background provided.
I would suggest a minor change in the title, as the sentence "...induces behavioral and neurotoxicity..." seems to be missing something.
Overall the language was clear and it was a very pleasant and interesting read.

Experimental design

The research question was well defined and the investigation rigorous.
The methods were described with sufficient detail, I would only add the volume of the solution that was being replaced daily (line 119), as well as the intensity of the light (or percentage of light intensity as given by the DanioVision chamber) used in the photomotor response, thigmotaxis and sensitization experiment.

Validity of the findings

No comment

Additional comments

No additional comments.

Reviewer 3 ·

Basic reporting

This study reported toxicity testing of fentanyl using zebrafish as a model organism. Fentanyl is a prescription drug often used during pregnancy, so it is of interest to understand its potential toxicity. However, the study failed to explain the rationale of the experimental design, especially on the choice of dosage and the exposure route selection. As a result, it is difficult to comprehend how would the results contribute to the understanding of the hazard potential of fentanyl.

Experimental design

As mentioned above, the experimental design is critical to put the data acquired into understanding the implications of fentanyl usage. Besides, it is also important to provide negative and positive controls in the experiments to really understand the toxicity level of fentanyl.

Validity of the findings

Without a proper justification, the data presented in this study has limited value.

Reviewer 4 ·

Basic reporting

[√] Clear, unambiguous, professional English language used throughout.
- Nothing apparent stood out to me except for Line 73 (neurotoxic -> neurotoxicity)

[√] Intro & background to show context literature well referenced & relevant.

[√] Structure conforms to Peerj standards, discipline norm, or improved for clarity.

[√]Figures are relevant, high quality, well labelled & described.
- Figures 2-7 do not report the “F statistic” for the ANOVA performed in the description, nor do they accurately report N/group used in each the statistical analysis.

[] Raw data supplied (not supplied) hard to determine the statistical validity if the data is not there (could not find)

Experimental design

[√] Original primary research within Scope of the journal.

[√] Research question well defined, relevant & meaningful. It is stated how the research fills an identified knowledge gap.

[√] Rigorous investigation performed to a high technical & ethical standard. (this is debatable, however, I let this pass. Specifically, technical standard is hard to interpret when the statistics are hard to interpret)

[√] Methods described with sufficient detail & information to replicate. (yes and no, reasons below)
1.) Raw Data not supplied:
For all behavioral experiments they would need to provide time bins for each individual subject for the total duration of the specified test to be considered “raw data”…… the excel sheet provided are averaged values and show omitted values with no explanation as to why.

2.) Statistical analysis:
- Not clear why a “Welch’s ANOVA” is used when all statistical significance reported and shown on figures is a one-way ANOVA.
- Differences in F statistic reported indicate data was omitted with no clear explanation as to why.
Results:
Fentanyl exposure resulted in decreased behavioral capacity: (FIG 2)
- Line 207 (Welch’s ANOVA)?
- why are (F(x,?) different?
- no die offs at all?....
- no reported N per group in fig 2
Fentanyl exposure resulted in increased anxious behavior:
- why do df within and not stay consistent?
Fentanyl exposure produced deficient social behavior:
Discussion:
- fig 2 and 6 .. dramatic decrease in movement of all treatment groups

Validity of the findings

[√] Impact and novelty not assessed… Meaningful replication encouraged where rational & benefit to literature is clearly stated.

[] All underlying data have been provided; they are robust, statistically sound & controlled. (I could not in good faith check this based upon figures and discussed interpretation of said figures… I would need to see raw data but, there is some undisclosed data that appears to be missing)

[1/2 a √] Conclusions are well stated, linked to original research question & limited to supporting results. (some conclusions are well supported, some not so much)

Additional comments

1.) Raw Data not supplied:
For all behavioral experiments they would need to provide time bins for each individual subject for the total duration of the specified test to be considered “raw data”…… the excel sheet provided are averaged values and show omitted values with no explanation as to why.

2.) Statistical analysis:
- Not clear why a “Welch’s ANOVA” is used when all statistical significance reported and shown on figures is a one-way ANOVA.
- Differences in F statistic reported indicate data was omitted with no clear explanation as to why.

3.) Behavioral sensitization:
- When comparing the “Distance Traveled” in Figure 2 (5dpf) to Figure 6 (14dpf) there is a noticeable decrease in the average distance moved in the control group… indicating the overall “health” of the fish is of concern at this time point.
- Additionally, unless these are the exact same fish tested at 5dpf this claim is not valid

4.) I commend the authors for the results of Figure 7 however, the mRNA expression levels should have been evaluated at all age points tested to fully connect the behavioral differences to a measurable neuronal change.

5.) Although the results are compelling, the data analysis should be improved by providing rational for why certain subjects had their data omitted and explanations for each behavioral test on how the data is broken down with relevance to the duration of the test.
Notes (while reading):
Abstract:
Article Focus: Excessive fentanyl exposure in developing organisms -> drug-related withdrawal symptoms
Methods: Fentanyl exposure 3 Conc. (0.1, 1, & 5 mg/L, [0-5dpf]) -> no fen (6-10dpf) -> behavior on day 10
-Evaluated 6 behavioral assays.
Results:
-2 Conc. Showed results for a change in 4 of 6 behavioral assays.
- Substantial alteration in genes related to neuronal development.
Main Claim: Fentanyl exposure during embryonic development is neurotoxic.

Introduction:
- Line 73 (neurotoxic -> neurotoxicity)
- Drug conc. rationale used for zFish was determined by reported blood concentrations in humans (dwyer et al 2017).

Methods:
Zebrafish maintenance and reproduction:
- Clarification on larval feeding needed (Line 106 – 109) … straw worm? fungus shrimp?
Zebrafish exposure to fentanyl:
- Line 121 now referred to as “grass worms”
Shoaling behavior:
- reported using all noldus products then jump to zebralab (viewpoint software)…this seems odd
ANIMAL USE:
150 embryos per concentration: there is no explanation as to where other embryos went
- PMR (5dpf,N=24/group) -> 126 remaining
- Thigmotaxis (10dpf, N=24/group) -> 102 remaining
- Light/dark preference test (10dpf, N=24/group) -> 78 remaining
- Shoaling behavior (11dpf, N=20/group) -> 58 remaining
- Mirror-attack (12dpf, N=12/group) -> 46 remaining
- Sensitization Behavior (14dpf, N=12/group) -> 34 remaining
- qPCR (5dpf, N=30/group) -> 4 remaining

Results:
Fentanyl exposure resulted in decreased behavioral capacity: (FIG 2)
- Line 207 (Welch’s ANOVA)?
- why are (F(x,?) different?
- no die offs at all?.... suspected data manipulation
- no reported N per group in fig 2
Fentanyl exposure resulted in increased anxious behavior:
- why do df within and not stay consistent?
Fentanyl exposure produced deficient social behavior:
Discussion:
- fig 2 and 6 .. dramatic decrease in movement of all treatment groups

---

## Round 0.2 · Minor Revisions

· Academic Editor

Minor Revisions

Dear authors, as you can see from the comments below, the Reviewers were mostly satisfied with the revised manuscript. However, Reviewer 3 is asking to consider adding some additional relevant references to the manuscript, to better reflect the current state of the art in using zebrafish for toxicity assessment of new drugs and compounds. Please consider this suggestion and if adding references could improve the manuscript.

Reviewer 1 ·

Basic reporting

No comment

Experimental design

No comment

Validity of the findings

No comment

Additional comments

The authors presented a satisfactorily rebuttal letter and I have no more comments to add to the previous revision.

Reviewer 3 ·

Basic reporting

The rationale is better but needs further improvement. Suggest to include a few current studies of zebrafish for toxicity assessment of new drugs and compounds to strengthen the importance of this line of study.

Experimental design

There are increasing studies using various behavioral endpoints of zebrafish to evaluate the effects of chemicals and pollutants. Although there might be different mechanisms behind the changes in behavior, similar effects on the endpoints are certainly available in the literature.

Validity of the findings

no further comments.

---

## Round 0.3 · accepted · Accept

· Academic Editor

Accept

Dear authors, thank you for addressing the Reviewers' comments and modifying the manuscript accordingly. The manuscript is now suitable for publication.